# Day length regulates gonadotrope proliferation and reproduction via an intra-pituitary pathway in the model vertebrate *Oryzias latipes*

Muhammad Rahmad Royan, Kjetil Hodne, Rasoul Nourizadeh-Lillabadi, Finn-Arne Weltzien [ID] , Christiaan Henkel [ID] & Romain Fontaine [ID] ✉

In seasonally breeding mammals and birds, the production of the hormones that regulate reproduction (gonadotropins) is controlled by a complex pituitary-brain-pituitary pathway. Indeed, the pituitary thyroid-stimulating hormone (TSH) regulates gonadotropin expression in pituitary gonadotropes, via *dio2*-expressing tanycytes, hypothalamic Kisspeptin, RFamide-related peptide, and gonadotropin-releasing hormone neurons. However, in fish, how seasonal environmental signals influence gonadotropins remains unclear. In addition, the seasonal regulation of gonadotrope (gonadotropin-producing cell) proliferation in the pituitary is, to the best of our knowledge, not elucidated in any vertebrate group. Here, we show that in the vertebrate model Japanese medaka (*Oryzias latipes*), a long day seasonally breeding fish, photoperiod (daylength) not only regulates hormone production by the gonadotropes but also their proliferation. We also reveal an intra-pituitary pathway that regulates gonadotrope cell number and hormone production. In this pathway, Tsh regulates gonadotropes via folliculostellate cells within the pituitary. This study suggests the existence of an alternative regulatory mechanism of seasonal gonadotropin production in fish.

Reproduction is the most important biological function for all living organisms. For many animals, synchronization of reproduction with favorable environmental conditions is essential to ensure food abundance and moderate climate when their offspring are born in the spring/summer season[1]. For instance, animals with a long gestation (sheep and goat) breed between early autumn and winter, so that the offspring are born in spring or summer when grasses are abundant[2]. In fish, synchronization of reproduction with favorable environmental conditions is also required. For instance, Atlantic salmon reproduce in the fall with their offspring hatching in spring when water temperatures are higher and food availability starts to increase[3].

In vertebrates, all physiological events associated with reproductive function are regulated by the brain-pituitary-gonad axis. In the brain, the hypothalamic gonadotropin-releasing hormone (GnRH) neurons integrate environmental and internal cues, and stimulate the synthesis and release of two gonadotropins (follicle-stimulating hormone, FSH; and luteinizing hormone, LH) produced by gonadotropes in the pituitary[4]. Gonadotropins

are then secreted into the bloodstream to stimulate gametogenesis and steroidogenesis in the gonads[5]. In addition to the hypothalamic control, gonadotropin production and release are also regulated by feedback signals from gonadal sex steroids, which act either directly on the gonadotropes or indirectly via the brain (e.g. the GnRH neurons)[6]. This system is highly conserved in vertebrates, except for a few differences at high taxonomic levels. For example, in fish, but not in mammals, Gnrh neurons directly innervate the anterior pituitary[7], and Fsh and Lh are produced by two distinct gonadotrope cell types[8,9]. These characteristics make teleost fish ideal models to investigate the differential regulation of the two gonadotropin hormones, as well as gonadotrope plasticity and development.

One of the seasonal cues, daylength (photoperiod), is considered to be a noise-free (an almost identical and predictable rhythm each year) and primary signal for the activation of the reproductive axis in most seasonal breeders[10,11]. In mammals, photoperiodic information received by the pineal gland in the brain fine-tunes the secretion of melatonin hormone, whose duration and amplitude regulate thyroid-stimulating hormone (TSH)

Department of Preclinical Science and Pathology, Faculty of Veterinary Medicine, Norwegian University of Life Sciences, Ås, Norway.
✉ e-mail: romain.fontaine@nmbu.no

production in the pituitary *pars tuberalis* (PT)[12]. This PT-TSH signal then regulates gonadotropin production in the pituitary *pars distalis* (PD) via a retrograde pathway in the brain involving tanycytes expressing type 2 deiodinase (Dio2) from the 3rd ventricle as well as hypothalamic Kisspeptin, RFamide-related peptide (RFRP3), and GnRH neurons[12–15]. In birds, the same pathway exists, except that PT-TSH is regulated by deep photo-receptors that directly sense photoperiod instead of melatonin[12,13].

Although photoperiod regulates reproduction and gonadotropin levels in several fish species (e.g. stinging catfish[16], damselfish[17], and honmoroko[18]), little is known about the pathways involved. Teleost fishes do not possess an anatomically distinct PT[11], but some have a structure below their hypothalamus called the *saccus vasculosus* (SV)[19]. The SV is proposed to process photoperiodic signals and to produce TSH in masu salmon[13,20], although its involvement in reproduction remains unclear. In addition, nothing is known about the pathway regulating gonadotropes in teleost species that do not possess the SV[19]. In Atlantic salmon (which possess a SV), the pituitary product of the *tshbb* gene has been proposed to be comparable to PT-TSH in mammals and birds[21]. *tshbb* is regulated by photoperiod[22], and has orthologs in many non-Salmonidae species, including zebrafish and medaka[23]. In addition, several melatonin receptor isoforms show photoperiod-dependent expression levels in the pituitary in several teleost species, including in medaka[24,25] and Atlantic salmon[26]. Nonetheless, evidence for the functional role of pituitary Tsh and melatonin receptors in teleost reproduction remains scant.

Finally, despite the photoperiodic effect on *fshb* and *lhb* levels, it is not clear whether this regulation is due to increased gene expression, increased cell number, or both. Indeed, previous studies suggested that changes in Fsh and Lh cell number sometimes occur in the fish pituitary, allowing a rapid increase of *fshb* and *lhb* levels when increased hormone production by existing cells is not sufficient[27–29]. While changes in pituitary gonadotrope cell number are usually explained by mitosis, transdifferentiation, differentiation of progenitor cells, and apoptosis[6,30], how the seasonal photoperiod signal controls gonadotrope cell number is not known.

In this study, we used Japanese medaka (*Oryzias latipes*), a long-day seasonal breeder, as a model to investigate how photoperiod affects reproductive functions and gonadotrope cell plasticity. The species is a powerful model for genetic and developmental studies[31–33] due to a wide range of resources, as well as genetic and molecular techniques. These include, for instance, the recently developed 3D atlas of the pituitary, which facilitates visualization of all endocrine cell populations[29], and transgenic lines in which endogenous *lhb* and *fshb* promoters control the synthesis of fluorescent reporter proteins[34,35]. Here, we take advantage of these tools to elucidate how photoperiod regulates gonadotrope proliferation. We first show that the activation of the reproductive function by the seasonal cue photoperiod is associated with not only increasing gonadotropin production via increased gene expression, but also via proliferation of pituitary gonadotropes, in the vertebrate model medaka. Second, we describe an alternative pathway to the pituitary TSH-brain *dio2*-expressing tanycytes pathway for the seasonal regulation of gonadotropes. This pathway, which also involves pituitary Tsh but remains intra-pituitary, uses a subset of pituitary folliculostellate cells as intermediate to reach gonadotropes.

## Results

### Long photoperiod stimulates reproduction and increases gonadotrope cell number

To evaluate the effect of photoperiod on reproductive capacity and gonadotrope cell number, we exposed adult fish to two different light regimes: Short (10 h of light, SP) and long (14 h of light, LP) photoperiods (Supplementary Fig. 1a). After 3 months in these conditions, we counted the percentage of spawning females, shown by oviposited eggs, over seven days. While females in SP did not spawn at all, between 30% and 50% of the females in LP spawned each day (Supplementary Fig. 1b, c), confirming that LP stimulates reproduction in medaka.

Consistent with the reproductive activity observed in fish kept in LP condition, both males and females kept in LP showed a significantly higher

gonadosomatic index (GSI) compared to those kept in SP (Supplementary Fig. 1d). This suggests that LP stimulates gonadal development, which is in line with the higher *fshb* and *lhb* levels observed in LP fish compared to SP fish ($p_{fshb, male}$ = 0.027; Supplementary Fig. 1e, f). In addition, the pituitaries from LP fish contain a significantly higher number of Fsh and Lh cells than pituitaries from SP fish ($p_{male, Fsh cells}$ = 0.007; Supplementary Fig. 1g). This indicates that LP not only stimulates gonadotropin mRNA production, but also gonadotrope cell proliferation, and suggests that higher hormone-encoding mRNA production observed in LP exposed fish is at least partly due to gonadotrope hyperplasia. Furthermore, the percentage of Fsh and Lh cells in the pituitary of LP fish is 2-fold higher than that of SP fish ($p_{male, Fsh cells}$ = 0.017; $p_{male, Lh cells}$ = 0.012; Supplementary Fig. 1h), which provides evidence for the importance of gonadotrope hyperplasia in pituitary plasticity as well as in reproduction under LP conditions.

We found that SP females did not reproduce at all when coupled with LP males. This interesting finding confirms that photoperiod length strongly regulates reproduction in female medaka (Supplementary Fig. 1i). However, despite the lower levels of *fshb* and *lhb*, number of gonadotropes, and GSI, SP males were still able to reproduce, as shown by 7 out of 10 couples of SP males and LP females reproducing with 10–100% fertilized eggs. These results suggest that males are less affected by photoperiod for the control of their reproductive activity. For this reason, we focused on females in the rest of the study, and males are analyzed separately in Supplementary Fig. 2.

### Changes in photoperiod induce rapid physiological modifications

To determine how quickly the physiological modifications occur following a change in photoperiod, we raised fish in either SP or LP for 4 months before switching them to LP or SP, respectively (Fig. 1a). After 14 days, SP raised fish exposed to LP (SPtoLP) and LP raised fish exposed to SP (LPtoSP) had started and stopped reproducing, respectively.

We then looked at gonadotropin mRNA levels (*lhb* and *fshb*) and observed that both decrease in LPtoSP females, but increase in SPtoLP females (Fig. 1b, c). In SPtoLP females, *fshb* levels are significantly higher at day 7 compared to day 0 (*p* = 0.010). In contrast, changes in *lhb* levels only tend towards difference after 14 days (*p* = 0.061). In LPtoSP females, both *lhb* and *fshb* significantly declined by day 14 ($p_{fshb}$ = 0.031; $p_{lhb}$ = 0.003). These major changes in gonadotropin levels after 14 days likely explain why the fish started or stopped reproducing at this time point. These changes are generally consistent with the observed changes in estradiol (E2) levels which significantly increased in 14 days in SPtoLP females (*p* = 0.009) and trended down in LPtoSP females (Fig. 1d). The levels of *cyp19a1b* (encoding aromatase, which is known to be expressed in medaka pituitary gonadotropes[28] and folliculostellate cells[36]) show the same pattern as the gonadotropins (Fig. 1e).

By contrast, in the brain, *gnrh1* (which encodes the primary stimulatory factor for Lh gonadotropes[37]) and *kiss1* levels significantly increase in SPtoLP ($p_{gnrh1}$ = 0.029; $p_{kiss1}$ = 0.029). *gnrh1* (but not *kiss1*) levels also reach close to significance in LPtoSP females ($p_{gnrh1}$ = 0.051; $p_{kiss1}$ = 0.634) (Fig. 1f, g). These results suggest that the biological response to photoperiod is therefore regulated either at another level than mRNA (hormone production and/or release) in the brain or downstream from the brain (e.g. in the pituitary).

### Long photoperiod stimulates gonadotrope proliferation via mitosis in females

Because the changes in *lhb* and *fshb* mRNA levels were found to significantly increase after increasing photoperiod from SP to LP, we investigated whether LP induced gonadotrope cell mitosis. We thus performed BrdU incorporation experiments to determine whether gonadotropes were proliferating (Fig. 2a). While we saw no change in mitotic cell number in the pituitary of LPtoSP fish 5 days after the light regime change, with a total of about 60 proliferative cells per pituitary (Supplementary Fig. 3), we found that the total number of mitotic cells, as well as the number of mitotic Fsh

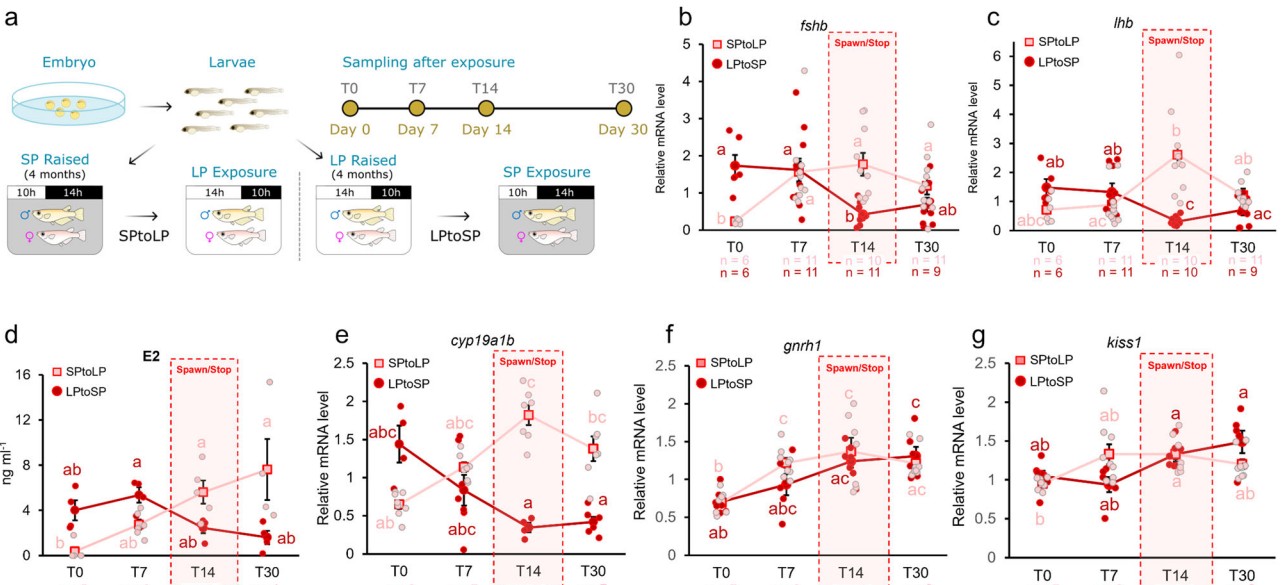

**Fig. 1 | Long photoperiod induces an increase in gene expression and circulating sex steroids. a** Illustration showing the experimental design. Fish were raised from hatching in LP or SP for 4 months before the photoperiod was changed. Fish were sampled on day 0 (just before the photoperiod change), 7, 14, and 30. **b–d** The fluctuation of pituitary *fshb* and *lhb* expression as well as blood E2 levels observed in females at different time points following the photoperiod change. **e–g** The fluctuation of *cyp19a1b* levels in the pituitary as well as *gnrh1* and *kiss1* in the brain at different time points following the photoperiod change. The letters (**a–c**) show statistically differences between each time points and group. The statistical analyses were performed using non-parametric tests, Scheirer–Ray–Hare Test, followed by Dunn's Test. All graphs are represented as mean ± SEM. Individual numbers (n) are annotated throughout the figure.

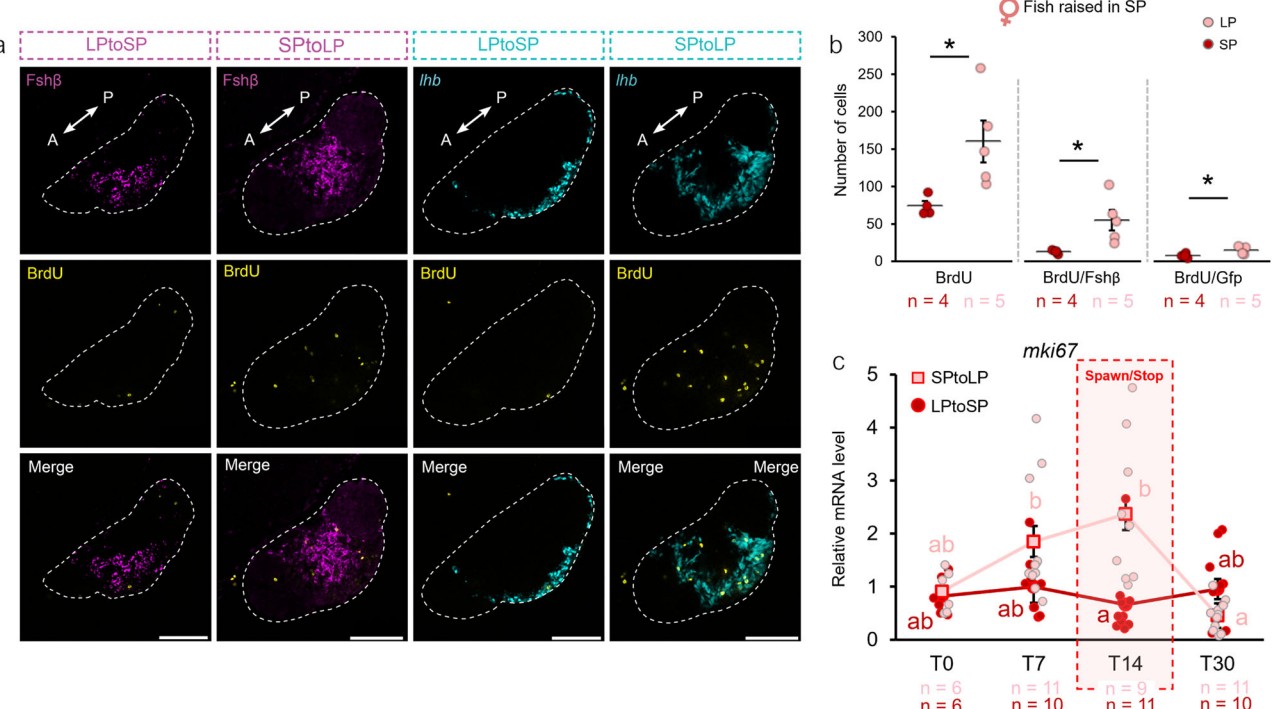

**Fig. 2 | Long photoperiod induces an increase in gonadotrope cell mitosis in SP females. a** Confocal planes illustrating Fsh (magenta) and Lh (cyan) cell mitosis using BrdU marker (yellow) 5 days after photoperiod change (LPtoSP and SPtoLP). Scale bar: 100 μm. The dashed line indicates the pituitary (A: anterior, P: posterior). **b** Number of mitotic cells in SP females after exposing them to LP for 5 days or kept in SP for control. **c** Relative expression levels of a mitotic cell marker, *mki67*, in females before and 7, 14 or 30 days following the photoperiod change as described in Fig. 1A. The statistical analyses were performed using two-sample independent t-test (**b**) and the non-parametric test Scheirer–Ray–Hare Test followed by Dunn's Test (**c**). The graph represents mean ± SEM. In **b** and **c**, the jittered dots represent each individual. In **b**, p-values < 0.05 are represented by *. In **c**, the letters (**a, b**) show statistically differences between each time points and group. Individual numbers (n) are annotated throughout the figure.

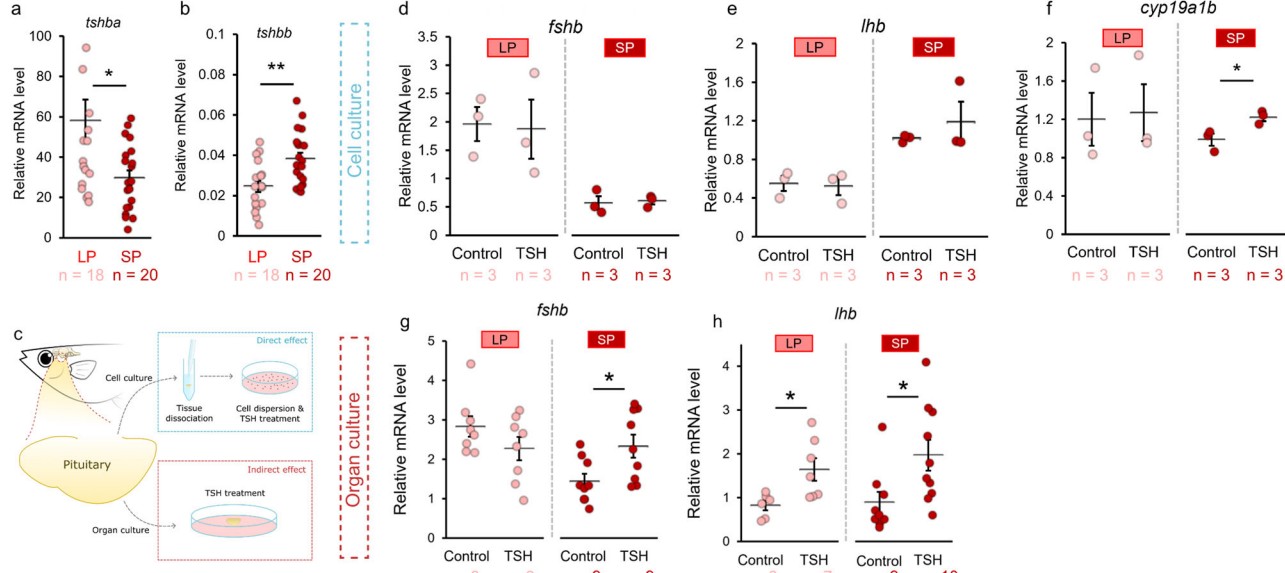

**Fig. 3 | Tsh indirectly stimulates gene expression of gonadotrope cells in SP females.** *tshba* (**a**) and *tshbb* (**b**) mRNA levels in LP and SP condition in female medaka. **c** Experimental scheme illustrating the treatment of 0.5 μM of bovine TSH pituitary extract to dissociated cell culture and organ culture of medaka pituitary to evaluate direct and indirect effect of Tsh on gene expression of gonadotrope cells. **d–f** Effect of TSH or vehicle (control) on *fshb*, *lhb* and aromatase (*cyp19a1b*) mRNA levels in dispersed pituitary cell cultures in female medaka from LP and SP condition (n = 3; in which each replicate represents 4 pooled pituitaries), the bar

represents mean ± SEM. **g–h** Effect of TSH or vehicle (control) on *fshb* and *lhb* mRNA levels in ex vivo medaka pituitary organ cultures of female medaka from LP and SP condition. The statistical analyses were performed using two-sample independent t-test (**b**, **d–f**) or Mann–Whitney U test (**a**, **g**, **h**). All graphs (unless otherwise stated) are represented as mean ± SEM with the jitter dots representing each individual (p-values: *< 0.05; **< 0.01). Individual numbers (n) are annotated throughout the figure.

and Lh cells significantly increased in SPtoLP females ($p_{total}$ = 0.031; $p_{Fsh\ cell}$ = 0.030; $p_{Lh\ cell}$ = 0.033; Fig. 2b). Together with the increased mRNA levels of the proliferation marker *mki67* observed in females at day 7 and day 14 (Fig. 2c), this clearly indicates that LP stimulates gonadotrope cell proliferation in females.

### Tsh indirectly regulates gene expression in gonadotrope cells

Since in mammals and birds the role of PT-TSH is crucial for the seasonal production of gonadotropins, we investigated whether Tsh also plays a role in regulating gonadotropin levels in medaka. Fish possess two paralog genes encoding proteins with similarity to TSHβ: *tshba* and *tshbb*[23], which both display photoperiod-dependent mRNA levels in females (Fig. 3a, b). While *tshba* levels were significantly higher in LP females than in SP females ($p$ = 0.030), *tshbb* levels were significantly lower ($p$ = 0.002; Fig. 3a, b). These results therefore are consistent with a role of both *tshba* and *tshbb* in the photoperiod signaling pathway in fish. Of note, the levels for *tshbb* were much lower than those of *tshba* as shown by the Cq values (average Cq values *tshba* = 19; *tshbb* = 32), as well as by pituitary RNA-seq where *tshbb* was not detected while high expression of *tshba* was found (Supplementary Fig. 4a). This suggests a minor role for *tshbb* compared with *tshba*, which is consistent with the higher homology between mammalian TSHβ and medaka Tshβa than between TSHβ and medaka Tshβb (Supplementary Fig. 4b–d). These results prompted us to further investigate the effects of Tsh on gonadotropin mRNA synthesis.

We applied bovine TSH pituitary extract, which has previously been demonstrated to activate fish Tsh receptors[38–40], to a dispersed pituitary cell culture (in vitro) in order to determine the direct effects on gonadotropes, and to pituitary organ cultures (ex vivo) to identify indirect effects (Fig. 3d–h). In dispersed pituitary cell cultures from females, we did not observe any change in *fshb* and *lhb* levels in any condition after TSH treatment (Fig. 3d, e). However, TSH stimulated *cyp19a1b* mRNA levels in cells from SP females ($p$ = 0.034; Fig. 3f). As paracrine effects among dispersed cells are reduced, these results suggest that some aromatase-expressing pituitary cells may be directly stimulated by TSH. Meanwhile, in

ex vivo pituitary organ culture from females kept under SP conditions, TSH significantly upregulated *fshb* ($p$ = 0.031) and *lhb* ($p$ = 0.024) levels (Fig. 3g, h) demonstrating that in females, Tsh regulates *fshb* and *lhb* gene expression in the pituitary indirectly.

### Tsh stimulates gonadotrope mitosis in females

Because long photoperiod stimulates both gonadotrope cell division and gonadotropin gene expression in females, we subsequently investigated whether Tsh regulates gonadotrope proliferation in pituitary organ cultures. First, bovine TSH downregulated *mki67* levels in LP ($p$ = 0.004) but upregulated levels in SP females (Fig. 4a). Next, we used BrdU treatment using pituitary from SP ($p$ = 0.034) females (as shown in Fig. 4b) and found that both the total number of mitotic cells and the number of mitotic Fsh cells increase after 12-h TSH treatment ($p_{total}$ = 0.034; $p_{Fsh\ cell}$ = 0.002; Fig. 4c, d). TSH treatment also elevated Lh cell mitosis close to statistical significance ($p$ = 0.057). These results suggest that Tsh stimulates both gene expression and mitosis of gonadotrope cells in females.

### Folliculostellate cells mediate Tsh regulation of gonadotrope cells

Considering the indirect regulation of Tsh on gene expression and proliferation of gonadotrope cells, we decided to investigate which pituitary cells might mediate the Tsh signals to gonadotrope cells.

Using the available pituitary bulk transcriptomic data (which have higher sequencing depth than single cell sequencing), we found that of the two Tsh receptor encoding genes identified in the medaka genome[23], only one is expressed above background level in the medaka pituitary (Supplementary Fig. 5a). This receptor does not show photoperiod-dependent expression levels (Supplementary Fig. 5b). Single cell pituitary transcriptomic data indicates that this Tsh receptor is specifically expressed in a small cluster of cells (Fig. 5a, b). This cell cluster also expresses common markers for non-endocrine folliculostellate (FS) cells (Supplementary Fig. 6), including high levels of aromatase (Fig. 5c). Using RNAscope, we localized Tsh-receptor expression in the dorsal part of the pituitary.

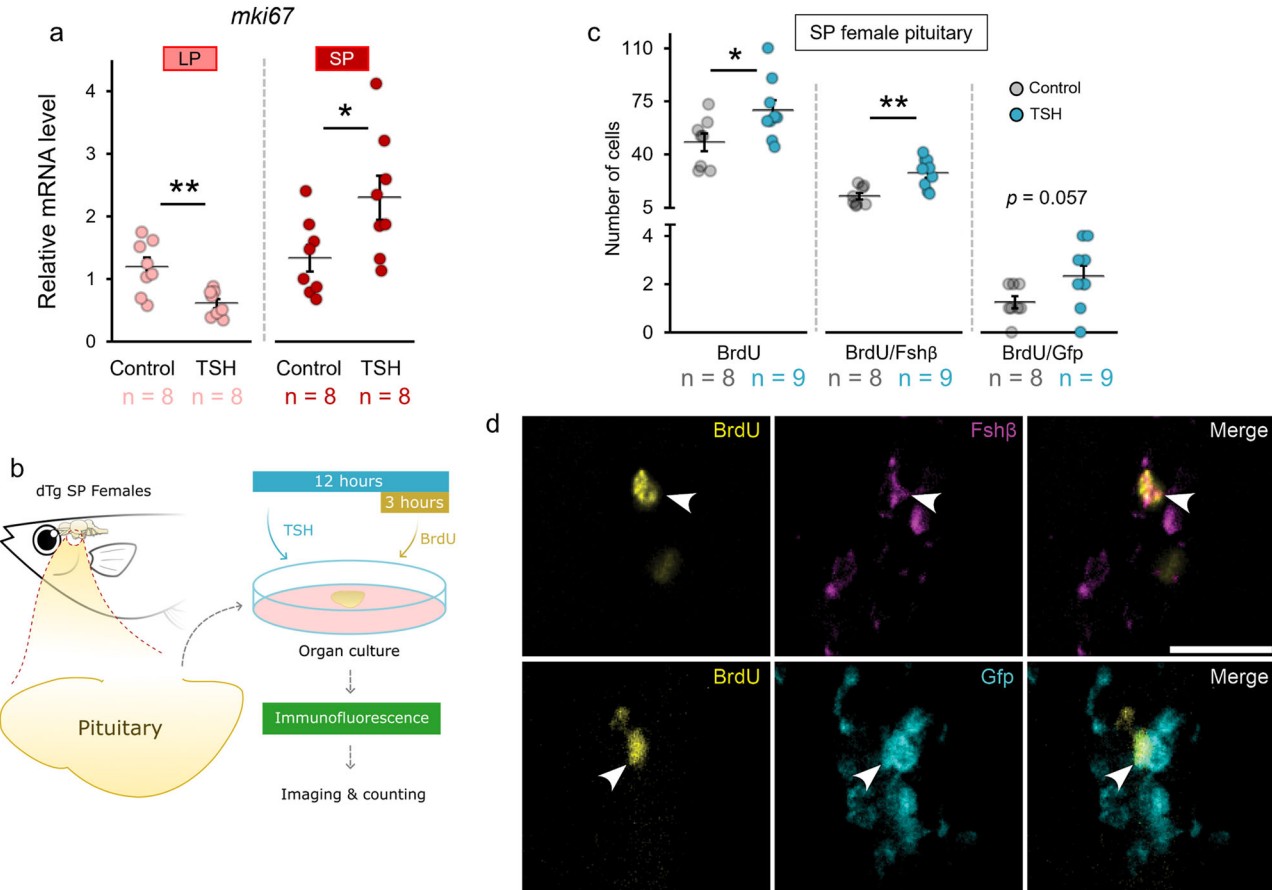

**Fig. 4 | Tsh stimulates mitotic activity in the pituitary of SP females. a** Effect of 0.5 μM of bovine TSH pituitary extract for 24 h on *mki67* levels in the female medaka pituitary organ cultures in LP and SP condition. **b** Schematic illustration of ex vivo medaka pituitary organ culture with bovine TSH pituitary extract and BrdU treatment to evaluate gonadotrope mitosis. **c** Number of mitotic cells in SP females exposed to TSH for 12 h. **d** Visual representation from the pituitary of dTg fish with Gfp signal for Lh cell (cyan), Fshβ for Fsh cells (magenta), and BrdU for mitotic cells (yellow) in the pituitary of SP females treated with TSH, as taken by Thunder fluorescence microscope. The statistical analyses were performed using two-sample independent t-test, in which the graph represents mean ± SEM while the jittered dots represent each individual (p-values: *< 0.05; **< 0.01). Individual numbers (n) are annotated throughout the figure. Arrows point to the mitotic cells. Scale bar: 20 μm.

Tsh-receptor expression labeling completely coincides with the labeling of cells expressing high levels of aromatase (Fig. 5d–f), thus confirming the single cell transcriptomic data. It also agrees with the increase of aromatase mRNA levels observed following TSH stimulation in dispersed pituitary cell cultures from females kept in SP condition (Fig. 3f). All these results thus support the direct regulation of FS cells by Tsh cells, suggesting that FS cells mediate the regulation of gene expression and mitosis of gonadotrope cells.

To further characterize the FS cells in the medaka pituitary, we labeled them using the commonly used dipeptide β-Ala-Lys-Nε-AMCA. FS cell bodies show a similar location distribution to the *cyp19a1b+/tshr+* cells (Fig. 6a, b). We observed that these FS cells send projections to gonadotropes in the ventral adenohypophysis (Fig. 6c–f), and that some FS cells are in close proximity to gonadotropes (Fig. 6g–j). Together these results suggest that the Tsh signal is relayed to gonadotropes directly by FS cells.

### Melatonin suppresses *tshba* gene expression in the medaka pituitary
We also investigated whether melatonin regulates Tsh production in medaka as it does in mammals. First, we confirmed previous results by showing that among the four melatonin receptors that have been described in medaka[24,25] (Supplementary Fig. 5c–f), two (*melr1a* and *melr1a-like*) show photoperiod-dependent expression levels, with higher levels in SP fish compared to LP fish (although only *melr1a-like* shows significantly different levels in the present study, $p_{male} = 0.034$; $p_{female} = 0.016$; Supplementary Fig. 5d).

We then investigated whether melatonin regulates *tshba* or *tshbb* expression in dissociated pituitary cell culture or in pituitary organ culture (Supplementary Fig. 7). In cell culture from female pituitaries, we did not observe any direct effect of melatonin treatment on either *tshba* or *tshbb* levels (Supplementary Fig. 7a, b). In pituitary organ culture, however, in agreement with a previous study where melatonin inhibited *tshba* levels in females[41], we found that melatonin significantly reduced *tshba* levels in males ($p = 0.020$; Supplementary Fig. 2q) and tended to decrease levels in females ($p = 0.093$) (Supplementary Fig. 7c). Contrary to *tshba* levels, *tshbb* expression levels were significantly upregulated by melatonin in pituitary organ cultures from LP females ($p = 0.001$; Supplementary Fig. 7d). Together, these results support a role for pituitary melatonin receptors in photoperiod signal integration in the pituitary through the regulation of Tsh synthesis and on gonadotrope cell proliferation.

### Discussion
While seasonal regulation of gonadotropin production in mammals and birds has been widely studied[12,13], this process is still poorly understood in fish. Using the teleost model organism the Japanese medaka, a species which reproduces in the wild during the summer when photoperiod is long and temperatures are warm[42], we investigated how photoperiod regulates gonadotropin mRNA production and gonadotrope cell number, as well as the regulatory pathways involved.

Following the confirmation in our laboratory that, as previously shown in refs. 42,43, medaka reproduction is induced by an increase in

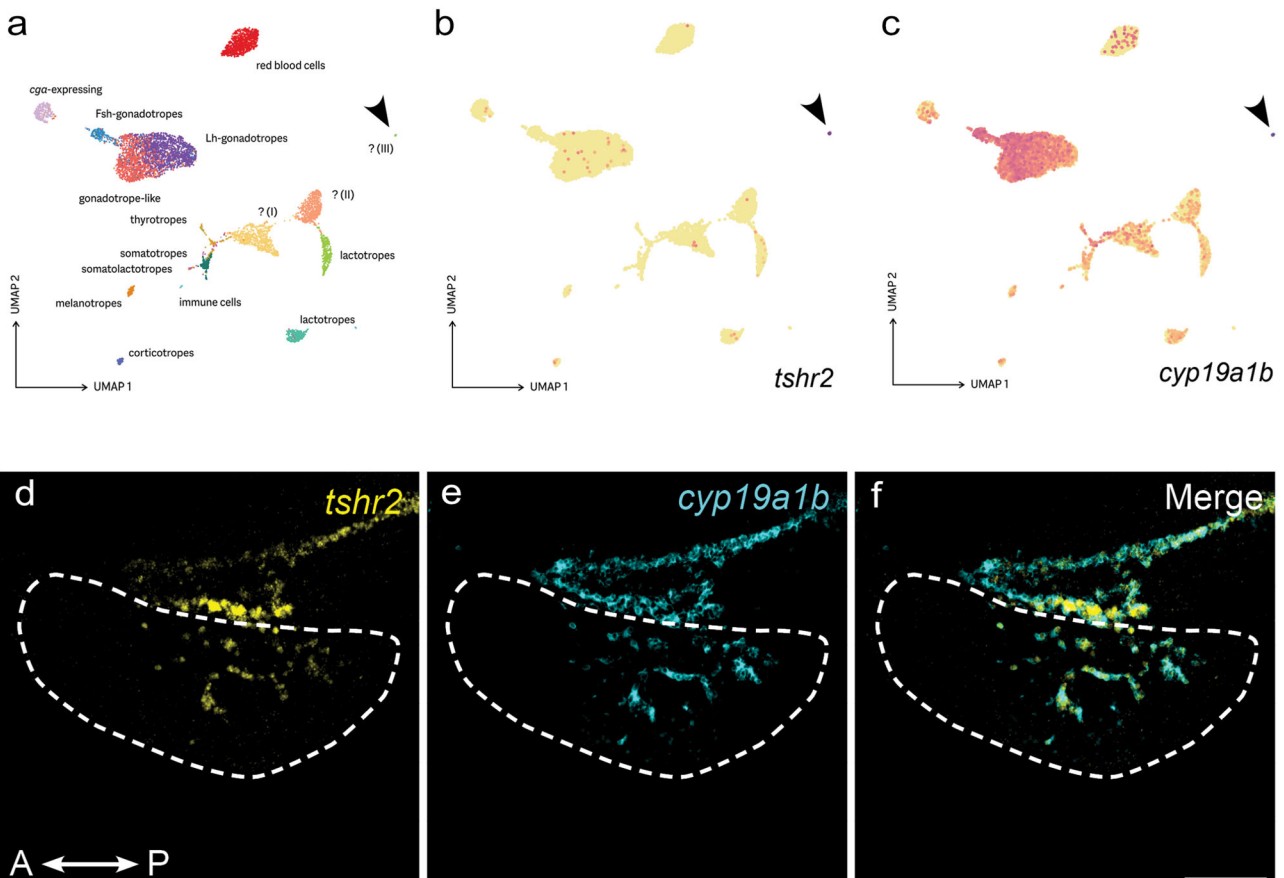

**Fig. 5 | Tsh regulation of gonadotrope cells is mediated via folliculostellate cells. a** The pituitary cell clusters identified by single-cell sequencing, showing three uncharacterized cell types (I–III). **b, c** *tshr2* and *cyp19a1b* are specifically expressed in one of these uncharacterized cell types (III), with *cyp19a1b* also showing modest expression in other pituitary cell types (e.g. gonadotropes). Figures are UMAP projections generated using Seurat. Color scale: log-transformed relative expression level (maximized per gene). **d–f** Confocal planes showing co-expression of *tshr2* and *cyp19a1b* in the same pituitary cells. Parasagittal section of brain and pituitary with a dashed line indicating the pituitary (A: anterior, P: posterior). Scale bar: 100 μm.

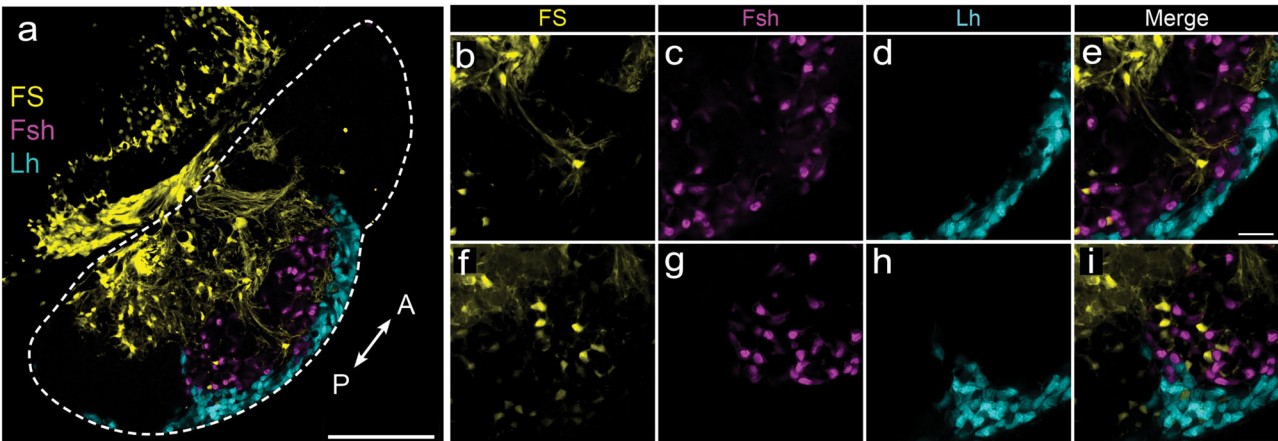

**Fig. 6 | Folliculostellate cells connect to gonadotropes. a** Labeling of folliculostellate cells using the dipeptide β-Ala-Lys-Nε-AMCA in double transgenic (*lhb*-GfpII/*fshb*-DsRed2) medaka fish. **b–e** High magnification of confocal planes showing the projection of folliculostellate cell extensions (yellow) to gonadotrope cells (Lh: cyan; Fsh: magenta; scale bar: 100 μm). **f–i** High magnification of confocal planes showing folliculostellate cells that border with gonadotropes. Dashed line indicates the whole parasagittal plane of the pituitary (A: anterior, P: posterior; scale bar: 20 μm).

photoperiod length, we found that gonadal development is stimulated under LP conditions via increased gonadotropin mRNA production in the pituitary. This agrees with observations in other teleost species[44–46], as well as in mammals[47]. In addition, we show here that the increase in gonadotropin transcripts is at least partly due to an increase of the gonadotrope cell number. In previous studies, we have indeed shown that changes in hormone production are often in line with changes in cell number[27–29,48]. Nevertheless, this is, to the best of our knowledge, the first

report of seasonal regulation of gonadotrope hyperplasia by photoperiod in any vertebrate.

Interestingly, while SP completely blocks reproduction in females, it does not in males, despite decreased *fshb* and *lhb* expression as well as lower Fsh and Lh cell numbers. Together with the fact that *fshb* or Fsh receptor (*fshr*) knockout medaka halt vitellogenesis in the female ovary, resulting in infertility due to immature follicles[49,50], our results support a pivotal role for gonadotropins in ovarian development and maturation in females. However, in agreement with a previous study in which male medaka produced mature spermatozoa even under SP conditions[42], spermatogenesis still occurs in SP medaka males in our study, even though fewer gametes or spermatozoa might be produced in those males, as indicated by the lower GSI compared to LP males. In addition, the fact that SP males are still able to reproduce suggests that spermatogenesis may not be completely dependent on gonadotropins, and that other factors could compensate for low gonadotropin levels. This hypothesis is consistent with previous knockout studies in male medaka, which demonstrated that the loss of *fshb* or *lhb*[49] or double knockout (KO) of Fsh and Lh receptors[50,51] does not disrupt spermatogenesis. In male zebrafish, although testis development is delayed significantly, Fsh-deficient fish are still fertile[52]. Lh can compensate for Fsh deficiency by activating the Fsh receptor in zebrafish[53,54], suggesting that low levels of either or both Fsh and Lh hormones could still, to some extent, stimulate spermatogenesis. Meanwhile, double KO of *fshb* and *lhb* in zebrafish males causes infertility[54], which suggests that the essentiality of gonadotropins depends on the species. Interestingly, in an in vitro experiment, human FSHR-expressing cells could be stimulated not only by human-FSH, but also by human-TSH, in a dose-dependent manner[55]. TSH was also shown to have a pronounced effect on the development of chicken testis[56]. As *tshba* levels increase in males (but not in females) in SP conditions, where the males still produce sperm, we hypothesize that *tshba*-derived protein (called Tsh throughout the rest of this discussion) compensates for the low Fsh and Lh levels in SP, thus stimulating spermatogenesis. On the other hand, despite being able to reproduce in winter photoperiod conditions, male medaka might require an additional winter factor, such as temperature, to completely inhibit their reproductive capacity[57].

Following the observation of gonadotrope hyperplasia in LP males and females, we have demonstrated to the best of our knowledge for the first time that photoperiod can regulate gonadotrope cell mitosis in vertebrates. Although stimulation of gonadotrope proliferation by photoperiod has never been shown before in any vertebrate, mitosis of gonadotrope cells has previously been observed in the medaka pituitary. In fact, our previous studies revealed an important role of gonadal sex steroids in the regulation of gonadotrope mitosis in the medaka pituitary[28,48]. Surprisingly, despite increased Fsh and Lh cell numbers in LP males compared to SP males, we found that LP actually inhibited gonadotrope mitosis in males. The difference between females and males could simply results from a sexual dimorphism in the timing of proliferation, as gonadotrope cell mitosis was only investigated at one time point (five days after the change in photoperiod). In ewes, it has been shown that cell division in the pituitary *pars tuberalis* decreased in short photoperiod, although the cell types were not identified[58]. By contrast, photoperiod change does not affect cell division in the pituitary PT in male sheep[59], suggesting that sex differences in cell division by the effect of photoperiod also occur in other vertebrate species. However, it is also possible that LP-induced gonadotrope cell hyperplasia results from another mechanism in males. Indeed, we previously demonstrated that Fsh cell hyperplasia originates from Fsh cell mitosis in both sexes, as well as transdifferentiation of Tsh cells in females, while a decreasing Sox2-immunolabeled cell population in males suggested that progenitor cells may contribute in gonadotrope proliferation in males[48]. Thus, it would not be surprising if males exposed to LP use another mechanism than mitosis, such as the differentiation of progenitor cells, to increase gonadotrope cell number.

In mammals and birds, *pars tuberalis* (PT)-TSH plays an essential role in regulating gonadotropin production indirectly via a retrograde pathway through the brain. Interestingly, *tshba* and *tshbb* levels are also photoperiod-dependent in medaka. The mRNA levels of *tshbb* are consistently upregulated in SP medaka, consistent with previous reports on stickleback[60]. This is in contrast to Atlantic salmon, where *tshbb*, which is strongly associated with smoltification[21], is upregulated in LP conditions[22]. For medaka as a non-smoltifying species, *tshbb* upregulation concurrently with *fshb* and *lhb* suppression in SP suggests that *tshbb*-expressing cells (further called Tsh-like cells throughout) might play an inhibitory role for reproduction, at least in females. Very little is known about Tshβb, apart from that the protein sequence is quite different than Tshβa. For instance, we do not know if it binds to the known Tsh receptors or to other types of receptors. Nevertheless, the extremely low levels compared to all other pituitary hormones suggest that Tsh-like might be used as a paracrine signal linked to photoperiodism, and not an endocrine signal.

We found that *tshba* is expressed at a much higher levels than *tshbb* in the medaka pituitary, which agrees with findings in Atlantic salmon[21]. Furthermore, the effect of photoperiod on *tshba* expression is sexually dimorphic in medaka with high levels in LP conditions in females and constant low levels in males. While sexual dimorphism in *tshba* levels has been previously reported in medaka[29,41], our findings support other fish studies where *tshba* was regulated by photoperiod (medaka[61], marine sticklebacks[60], and chub mackerel[62]). Previous studies also suggested that *tshba* might play a role in gonadotropin regulation as for instance in chub mackerel in which *tshba, fshb* and *lhb* levels are upregulated in LP fish of both sexes[62], or in medaka in which thyrotropes cells have been shown to start to produce *fshb* after removal of sex steroid feedback[48]. Therefore, the fact that *tshba* is controlled by photoperiod, and that its mammalian homolog protein TSH regulates both gonadotropin gene expression and gonadotrope mitosis, further support its role in the seasonal regulation of gonadotropin synthesis.

Interestingly, in both males and females, the stimulatory effects of mammalian TSH on gonadotropin mRNA levels were observed on ex vivo isolated pituitaries, but not in dissociated cells. We cannot rule out that the different medium used in our in vitro and ex vivo cultures, or phenotypic conversion following cell dissociation[27], had an effect on the behavior of some cells. However, the main difference between the dissociated cell culture and the ex vivo culture is the lack of a structural network. In addition, while paracrine signals are still present in dissociated cell culture[63], they are also reduced compared to in ex vivo tissue culture. Therefore, our results suggest the existence of an indirect, intra-pituitary pathway for the regulation of gonadotropin mRNA production by Tsh.

The only pituitary cells expressing Tsh receptor genes are a subset of putative folliculo-stellate (FS) cells. These FS cells are therefore most likely the intermediate between Tsh cells and gonadotropes. Our single-cell data clearly shows that what are typically considered to be FS cells, represent in fact a heterogenous cell population, as the established marker genes (e.g. *s100*) are expressed in at least three distinct groups. As FS marker genes have been identified based on bulk analysis of β-Ala-Lys-Nε-AMCA-stained cells, this complex presumably also includes pituicytes[64,65]. Only one small cell cluster in this complex expresses the Tsh receptor, as well as high levels of aromatase. Our study therefore provides a starting point for further characterizations and functional analyses of FS/pituicyte heterogeneity. Interestingly, pituicytes are glial cells; in the mammalian brain, PT-TSH regulates another type of glial cells, the tanycytes[66]. In the brain of Atlantic salmon, *dio2* levels in the brain are correlated with pituitary Tsh levels[22], suggesting that the TSH-tanycyte *dio2* pathway also exists in fish. Therefore, the pathway that we describe in this study could then be an intra-pituitary variation on the TSH-tanycyte *dio2* path.

Expression of Tsh receptors has also been shown in FS cells from human[67,68], mouse[69], and chicken pituitary[70], suggesting that this intra-pituitary pathway for Tsh regulation of gonadotrope cells, mediated by FS cells, might be conserved among vertebrates. FS cells possessing cytoplasmic extensions have been suggested to assist in paracrine communications

**Fig. 7 | Schema of the proposed hypothesis on the photoperiodic regulation of gonadotrope cell mitosis via melatoinin, Tsh, and folliculostellate cells.** In summer photoperiod (right panel), melatonin levels are suppressed, allowing Tsh cells to stimulate gene expression and mitosis of gonadotrope cells, via folliculostellate cells. As a result, an increasing number of gonadotrope cell due to mitosis participates in the increase in gonadotropin production necessary for gametogenesis and steroidogenesis, which allow the fish to reproduce. In winter photoperiod condition (left panel), high melatonin levels indirectly suppress *tshba* expression. In absence of Tsh stimulation, gene expression and mitosis of gonadotrope cells remain low. By contrast, melatonin indirectly upregulates *tshbb* expression that might inhibit Fsh and Lh, but this needs to be confirmed.

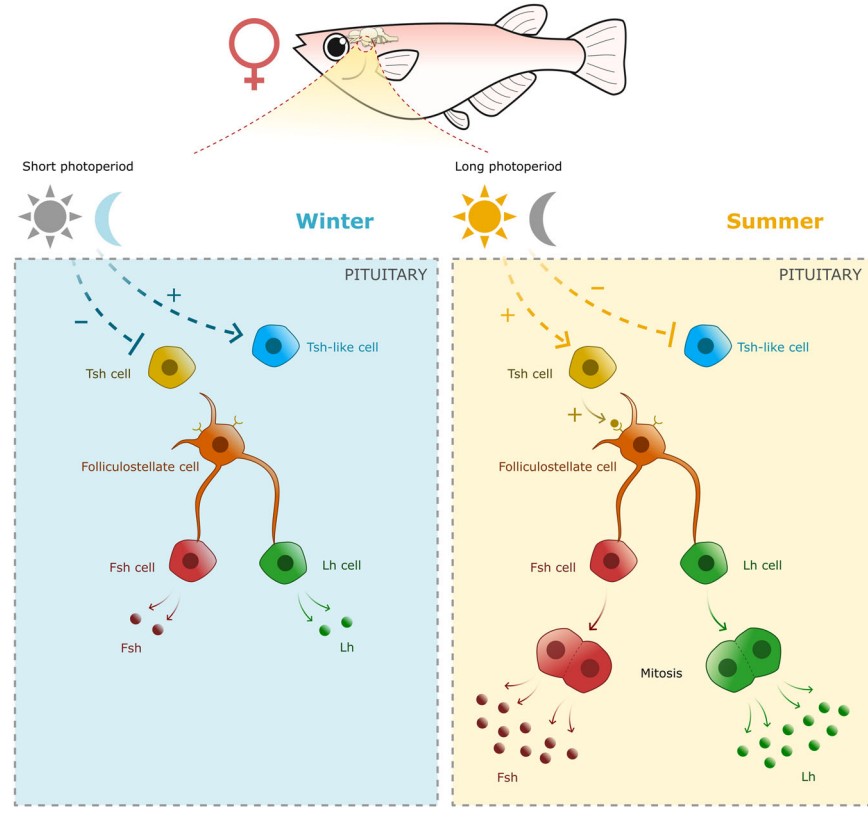

among pituitary cells[71–74]. The previously described gap junction-mediated calcium ($Ca^{2+}$) propagation among FS cells[72,75], as well as gap junction-mediated communication between FS and other endocrine cells[76,77], may enable a signal transduction cascade that induces upregulation of gene expression and mitosis in gonadotrope cells.

Finally, we show that, like in mammals and in agreement with previous studies[24], pituitary melatonin receptors may play a role in seasonal photoperiodism in medaka as they are differentially regulated by photoperiod. The peak of melatonin receptor expression is usually associated with the breeding season[11]. This is in apparent contrast with our result where melatonin receptor expression is upregulated under non-breeding conditions. However, this is consistent with a previous study in which most melatonin receptors show daily cycles and tend to be higher in SP than in LP[24], as well as elevated at night under breeding conditions[25]. The elevated melatonin receptor during SP in medaka may facilitate the inhibition of Tsh signaling to gonadotrope, thus blocking reproduction. In particular, melatonin seems to downregulate *tshba* in females, supporting a previous study where melatonin was found to significantly decrease *tshba* levels in females[41]. This is intriguing as SP instead increases *tshba* levels in males, suggesting that another factor, more potent than melatonin, is regulating *tshba* levels in males.

The identity of the cells expressing melatonin receptors remains to be determined in fish. Nevertheless, because *tshba* regulation by melatonin occurs only in ex vivo pituitary cultures and not in dispersed pituitary cell culture, our results suggest that contrary to in mammals, melatonin receptors are expressed by another cell type than thyrotropes. This unknown cell type would thus play the intermediate role between melatonin signal and thyrotropes for the regulation of *tshba* synthesis. Nevertheless, the role of melatonin in regulation of seasonal reproduction still remains unclear. Even if our study suggests a link between melatonin receptors and Tsh regulation, there are several studies in different fish species questioning the role of melatonin in controlling the reproductive function[78]. Whether its importance is species-dependent remains to be determined. On the other hand, deep photoreceptors have been found in all the tissues that have been

investigated in fish, including the pituitary[79,80]. A recent study showed that deep photoreceptors such as opsin play a role in pituitary hormone release[81]. However, whether deep photoreceptors play a role in Tsh regulation as in birds is still unknown.

In summary, because gametogenesis still occurs in SP males the role of *tshba*, FS cells, and melatonin in the regulation of gene expression and proliferation of gonadotrope cells is less clear in males than in females. It is possible that in males, another environmental factor such as temperature plays a stronger or concomitant role in the seasonal regulation of gonadotropes. In female medaka, we propose the following hypothesis (Fig. 7): In summer conditions (long photoperiod), activation of Tsh cells stimulates gene expression and mitosis of gonadotrope cells via FS cells. The increase of gonadotrope cell mitosis increases gonadotrope cell number, which contributes to the elevation of gonadotropins necessary for gonadal development and maturation, and thus for the fish to reproduce.

## Methods
### Experimental animals
Male and female of wild type (WT) and double transgenic (dTg) (*lhb*:hrGfpII/*fshb*:DsRed2) line[35] medaka (Hd-rR strain) were raised at 28 °C in a re-circulating water system (pH 7.5; 800 µS) with 14 h light and 10 h dark (long photoperiod, LP) or 10 h light and 14 h dark (short photoperiod, SP). The photoperiod regimes were based on what has been previously performed in medaka to mimic seasonal variation[82] and the light source used was the regular LED supplied by Pentair on (490–633 lux depending on where the sensor is placed in the box; for color spectrum see Supplementary Fig. 8). Fish were fed three times daily using artemia (twice) and artificial feed (once). The sex was determined based on secondary sexual characteristics[83].

### Ethics approval
Experiments with live vertebrates performed in this study were approved by the Norwegian Food Safety Authorities (Permit #24305).

**Table 1 | Primary and secondary antibodies used for immunofluorescence**

| Antibody | Dilution | Source | Reference |
|---|---|---|---|
| Rat anti-BrdU | 1 : 250 | abcam (ab6326) | [28] |
| Rabbit anti-medakaFshβ | 1 : 500 | custom-made | [85] |
| Goat anti-rabbit (Alexa-555) | 1 : 500 | Invitrogen (A21429) | [35] |
| Donkey anti-rat (Alexa-647) | 1 : 500 | Invitrogen (A78947) | [28] |

**Photoperiod exposure in vivo**

*Experiment 1*: To compare *fshb* and *lhb* gene expression and number of gonadotrope cells between LP and SP fish, 2-month-old medaka were distributed into two groups of mixed sex. The first group was kept in LP while the second group was kept in SP for 3 months before sampling. The number of females showing oviposited eggs was used as a measure of the percentage of spawning animals. *Experiment 2*: To address whether SP fish do not reproduce, we set up ten adult couples of SP males with LP females, as well as SP females with LP males. *Experiment 3*: To investigate the effect of photoperiod exposure on gene expression, fish were raised for 4 months in LP and exposed to SP, while those raised in SP for 4 months were exposed to LP, and sampled at day 0 (before exposure), 7, 14, and 30. As controls, the fish raised either in LP or in SP stayed in their original photoperiod condition during the trial period.

**Euthanasia, morphometric measurements, and gonadosomatic index (GSI) calculation**

The fish were euthanized by ice water immersion before the total body weight (mg) and the standard length (cm) were measured. Gonad weight (mg) was measured to calculate gonadosomatic index (GSI) using the following formula: GSI = Gonad weight/Total body weight * 100.

**Labeling of mitotic cells**

To investigate the photoperiodic effect on gonadotrope mitosis, dTg fish that underwent *Experiment 3* were treated at day five with 1 mM BrdU (Sigma) diluted in water with 0.33% dimethyl sulfoxide (DMSO) for 4 h. Immunofluorescence was performed as previously described[84] with minor modification as described below. After an overnight fixation at 4 °C in 4% paraformaldehyde (PFA; Electron Microscopy 135 Sciences) in phosphate buffered saline with Tween (PBST: PBS, 0.1%; Tween-20), the brain and pituitary complex from dTg fish were washed three times with PBST, embedded in 3% agarose ($H_2O$), and para-sagittally sectioned with 80 μm thickness using a vibratome (Leica). The free-floating sections were treated with 2 M hydrogen chloride (HCl) at 37 °C for 1 h for epitope retrieval before a 1-h incubation at RT in blocking solution (3% normal goat serum (NGS); 0.5% Triton; 1% DMSO in PBST). The tissue slices were then incubated at 4 °C overnight with previously verified anti-BrdU antibody (1:250; abcam; ab6326) and anti-medakaFshβ antibody (1:500; homemade[85]). After extensive washes with PBST, the slices were incubated for 4 h at RT with secondary antibodies (Table 1).

**Melatonin and Tsh effects on dispersed pituitary cell culture (in vitro) and pituitary organ culture (ex vivo)**

To investigate the direct effect of melatonin or Tsh, an in vitro dispersed pituitary cell culture experiment was prepared. Cells from four pituitaries from adult WT fish were were mechanically dissociated[86] using a glass pipette after a 30-min digestion in trypsin (2 mg/ml; Sigma), followed by a 20-min incubation in trypsin inhibitor (1 mg/ml; Sigma) and Dnase I type IV (2 μg/ml; Sigma). After centrifugation at $200 \times g$, the cells were resuspended in 200 μl growth medium (L-15; Life Technologies) adjusted to 280–290 mOsm with mannitol and pH 7.75 with 1.8 mM glucose, 10 mM $NaHCO_3$, and penicillin/streptomycin (50 U/ml; Lonza). The cells were plated in a 48-well plastic plate (Sarstedt) coated with poly-L-lysine (Sigma). The plate was prepared by adding 50 μl poly-L-lysine, left for 1 min before decanting the solution and washed with 500 μl MQ water, and air drying the

coated wells in a UV-light laminar flow hood for approximately 30 min. The experiment was run in triplicate.

To investigate the indirect effect of melatonin or Tsh, an ex vivo isolated pituitary organ culture was performed by detaching pituitaries from the brain and culturing three pituitaries together in growth medium supplemented with 5% fetal bovine serum (Sigma) as previously described[87]. Quickly after pituitary sampling (ex vivo) or pituitary cell dissociation (in vitro), the medium for all experiments above was supplemented with 10 μM of melatonin diluted in absolute ethanol (Sigma) or 0.5 μM bovine TSH pituitary extract diluted in water (Sigma) or vehicle alone, and incubated for 24 h at 26 °C and 1% $CO_2$. To limit the effects of phenotypic changes previously observed in dispersed pituitary cell culture[27], the cells/tissues were immediately collected and processed for RNA extraction after the end of the treatment. The TSH effect on gonadotrope cell mitosis was also evaluated using pituitary organ culture treated with 12 h of TSH and 3 h of BrdU, as explained above, before immunofluorescence (see Fig. 3C for illustration).

**Sex steroid extraction and enzyme-linked immunosorbent assays (ELISA)**

Blood was collected as previously performed[88], from the caudal vein using a glass needle (outer diameter 1 mm, inner diameter 0.5 mm; GD-1; Narishige) coated with 0.05 U/μl heparin sodium (Sigma) in phosphate buffered saline (PBS). The blood was stored at −80 °C until use. Sex steroid extraction and ELISA were performed as previously described[89], and the E2 and 11-KT levels were measured using ELISA kits (Cayman Chemical) according to the manufacturer's instructions. Briefly, the collected blood was diluted in PBS (1:200). Then, the steroid was extracted 3 times by vortexing the diluted blood with 4 times volume of diethyl ether (Sigma) followed by a flash spin. The organic layer was transferred into a clean tube, evaporated at 45 °C, and resuspended in ELISA buffer provided by the kit. Sex steroid concentrations were calculated with a standard curve fitted by a 4-parameter-logistic regression ($R^2 > 0.99$).

**Bulk and single-cell (sc) pituitary transcriptomics analysis (RNA-seq)**

We used an available RNA-seq dataset[90] from 16 male and 68 female medaka[91]. Read counts per gene were normalized using the total amount of reads (counts per million).

For the scRNA-seq, we used an available dataset[92] from 23 male and 24 female medaka[93]. We used the Seurat (version 3.2.3, R version 4.0.3) function FindMarkers to determine which genes are specific for the cell cluster expressing *tshr* and aromatase. Using the default Wilcoxon rank sum test for genes expressed in at least two cells in that cluster relative to all other cells, and showing a minimum log fold change of 1, we found 90 genes with an adjusted *p*-value (Bonferroni corrected) of <0.05. Several of these genes are known as FS markers from literature on other teleost species.

**Quantification of mRNA levels with quantitative polymerase chain reaction (qPCR)**

For in vivo and ex vivo experiments, the pituitary was collected and stored at −80 °C in 300 μl of TRIzol (Invitrogen) with 6–7 zirconium oxide beads (Bertin Technologies). For in vitro experiments, 300 μl of TRIzol was added to the wells, which were scraped using pipette tips before transfer to the tubes and storage at −80 °C until use. The samples were homogenized and mixed with 120 μl chloroform, and the pellet was reconstituted with 14 μl nuclease free water. cDNA synthesis was performed with a minimum of 120 ng of RNA using SuperScript III Reverse Transcriptase (Invitrogen) and random hexamer primers (Thermofisher Scientific). cDNA samples (5× diluted) were analyzed in duplicate, using 3 μl of the cDNA and 5 μM each of forward and reverse primer in a total volume of 10 μl (Table 2). The parameter cycle was 10 min pre-incubation at 95 °C, followed by 42 cycles of 95 °C for 10 s, 60 °C for 10 s and 72 °C for 6 s, followed by melting curve analysis. The mRNA level was normalized using *gapdh* and *rpl7* as reference

**Table 2 | Primer sequences used for mRNA level analysis in the medaka pituitary**

| Gene | Sequence (5' - 3') | Accession Number | PCR product size (bp) | Efficiency | Reference |
|------|-------------------|------------------|----------------------|-----------|-----------|
| rpl7 | F: TGCTTTGGTGGAGAAAGCTC | NM_001104870 | 98 | 2.03 | [85] |
| | R: TGGCAGGCTTGAAGTTCTTT | | | | |
| gapdh | F: CCTCCATCTTTGATGCTGGT | XM_004077972.3 | 170 | 2.01 | [85] |
| | R: ACGGTTGCTGTAGCCAAACT | | | | |
| lhb | F: CCACTGCCTTACCAAGGACC | NM_001137653.2 | 100 | 2 | [34] |
| | R: AGGAAGCTCAAATGTCTTGTAG | | | | |
| fshb | F: GACGGTGCTACCATGAGGAT | NM_001309017.1 | 73 | 2.03 | [85] |
| | R: TCCCCACTGCAGATCTTTTC | | | | |
| tshba | F: ATGTGGAGAAGCCAGAATGC | XM_004068796.4 | 88 | 2 | [29] |
| | R: CTCATGTTGCTGTCCCTTGA | | | | |
| tshbb | F: CCCAGTGCATTGCTATCAACAC | XM_011477157.2 | 79 | 2 | This study |
| | R: CCAAACCGGCCTCTAAAATTGG | | | | |
| mki67 | F: ACCAATCTGAGCACAGCCAAC | XM_011487928.3 | – | 2 | [99] |
| | R: GGTGCAGGTGGATACTCAAAC | | | | |
| cyp19a1b | F: AAGAAGATGATCCAGCAAGAG | XM_011475856.1 | 88 | 2 | [99] |
| | R: AGCATCAGAAGAAGTAAGAAAAGTG | | | | |

genes. The efficiency (E) value is calculated using the formula ($E = 10^{-1/slope}$), in which the slope of the regression line is based on a standard curve made by increasing diluted cDNA and the corresponding Cq values.

### Identification of the cells expressing Tsh receptors with RNAscope
After euthanasia, the blood was removed by cardiac perfusion[94] with 4% PFA (PBS). Brain and pituitary were dissected and fixed overnight at 4 °C with 4% PFA (PBST). Tissues were then incubated in 25% sucrose solution (diluted in PBS) overnight at 4 °C and mounted in a block with OCT (Tissue-Tek, Sakura) and stored at −80 °C until use. Tissues were later parasagittally sectioned with a CM3050 Leica cryostat (10 μm sections). RNAscope fluorescent multiplex V2 assay[95] (ACDbio) was carried out as described by the supplier and previously performed on medaka tissue[96], using tshr2 (ENSORLG00000014222) and cyp19a1b (ENSORLG00000005548) probes, and combined with opals 520 (tshr2) and 690 (cyp19a1b) (Akoya Bioscience) to avoid signal crosstalk during imaging.

### Labeling of folliculostellate cells
Pituitaries from adult dTg fish were directly incubated at RT in the dark for 4 h in 100 μM dipeptide β-Ala-Lys-Nε-AMCA (US Biological Life Sciences)[65] diluted in Ca²⁺-free extracellular solution (2.9 mM KCl, 134 mM NaCl, 1.2 mM MgCl₂, 4.5 mM glucose, and 10 mM N-2-hydroxyethylpiperazine-N-2-ethane sulfonic acid (HEPES), adjusted to osmolality 280–290 mOsm with mannitol and pH 7.75 with 1 M NaOH) with 0.5% bovine serum albumin. The pituitaries were then fixed in 4% PFA overnight and washed in PBST three times for 10 min at RT and incubated two days in PBST at 4 °C before mounting.

### Image processing and cell counting
Fluorescent images were obtained using a Leica Confocal Microscope (Dmi8, Leica) with 20× apochromat objective (numerical aperture 0.75), with laser wavelength 488 (Alexa-488, Opal 520), 555 (Alexa-555), 647 (Alexa-647, Opal 690). To avoid cross talk between fluorophores, the channels were obtained sequentially. Las X (v3.7, Leica) and ImageJ (1.53t; http://rsbweb.nih.gov/ij/) were used to process the images. Automatic cell counting was performed on selected z-sections (spaced by 1 cell diameter) using Cell Profiler software (v2.1.0[97]) as previously performed[28] while triple-labeled cells were counted manually using an ImageJ-based cell-counter plugin.

### Statistics and reproducibility
The sample size (3 at minimum) was based on at least the required minimum sample size for statistical analysis. No data were excluded from the analysis. All data generated were replicated at least once and were reproducible, except for the qPCR for gnrh1 and kiss1 (Fig. 1F, G). All the experimental animals that were in holding tanks were distributed randomly into specified groups. Sample collection was performed using systematic randomization. Collected data were tested for normality and homogeneity with Shapiro–Wilk and Levene's tests, respectively. Non-parametric tests were used for the data that did not pass with the normality or homogeneity test. Both the statistical test used and the number of independent samples (n) are described in the figures and their legends. Statistical significance was set to $p < 0.05$ and the tests used are described in figure legends. All sample groups were independent and all statistical analyses were performed using Jamovi (Version 2.2.5)[98]. The graphs are provided as mean ± standard error of the mean (SEM).

### Reporting summary
Further information on research design is available in the Nature Portfolio Reporting Summary linked to this article.

## Data availability
Source data for graphs and charts can be found in the Supplementary data file. Processed single-cell and regular RNA-seq data were retrieved from https://identifiers.org/geo:GSE162787 and https://doi.org/10.18710/HTCXRN, respectively. All other data are available from the corresponding author on reasonable request.

## Code availability
Data were processed in R (version 4.0.3) using the code available at Zenodo, https://doi.org/10.5281/zenodo.10729514.

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

## Acknowledgements
We thank Anthony Peltier, Lourdes Carreon G Tan, and Arturas Kavaliauskis for fish facility maintenance. We also than Prof. Dianne Baker for the English language editing. This work was funded by the Norwegian University of Life Sciences (R.F.), the Research Council of Norway (grant numbers 244461 and 243811, F.A.W.), and Fiskeri - og havbruksnæringens forskningsfond (grant number 901590, K.H.).

## Author contributions
M.R.R., R.N.L., R.F., C.H. did experimental work. M.R.R., C.H., R.F., F.A.W., K.H. designed the study. M.R.R., C.H., R.F., K.H. analyzed the data. M.R.R. wrote the first draft. M.R.R and R.F wrote the manuscript with inputs from all other authors.

## Competing interests
The authors declare no competing interests.
