## [Peer review file · Communications Biology]

Reviewers' comments:

Reviewer #1 (Remarks to the Author):

In this manuscript, Royan and co-authors describe an intra-pituitary paracrine circuit by which thyroid-stimulating hormone (TSH) stimulates gonadotroph cells indirectly through folliculostellate/pituicyte cells. By showing that TSH expression is affected by photoperiod (perhaps by melatonin) they claim that the aforementioned circuit is responsible for the photoperiod-dependent control of reproduction in this species.

The experimental data is very well presented and the TSH-FS-gonadotroph circuit is an interesting and novel finding in fish. However, the link between photoperiod and the circuit is less convincing and based too strongly on correlative observations.

Specific comments:

Line 42-44: The suggestion that the same mechanism applies to other vertebrates cannot be based just on the presence of the components in other species. Moreover, the role of seasonal TSH signaling through glial cells was described in mammals (PMID: 35361916) so the authors should perhaps rephrase their claim to state that the mechanism also exists in fish.

Throughout the study, the authors should not use the term cell "activity" as it may be misleading. Activity usually refers to electrical activity which they do not measure. Gene expression is a more appropriate term in this case since this is the parameter they use as a readout.

Authors should consider including PMID: 21093265 in their intro/discussion as it describes photoperiodic sensitivity of tsh genes in fish.

Line 125: Consider revising "the double" into 2-fold higher.

Line 125: Is support for the importance of FSH and LH in reproduction required? Perhaps it provides further evidence for the importance of hyperplasia in the gland's plasticity?

Lines 151-156: The paragraph makes a statement (gnrh1 and kiss signaling are elevated in SP toLP transition) and then counters it in the second part (the genes are also elevated in the opposite shift). It is unclear what the readers should make of this.

The authors should provide proof or at least cite papers that prove that bovine TSH activates the fish receptors. Considering the low homology of the bovine and medaka proteins (Supp. Figure 4) it is not a trivial question.

Line 189: Why is this evidence for direct activation of aromatase-expressing cells? Can they not be activated in a paracrine manner by other cells? Gonadotrophs express aromatase and therefore they can be the source of its rise. Is there an increase of E2 in the medium? That could be a way to induce at least LH expression.

Is the medaka TSHR2 activated by TSHba or TSHbb? Or both? This seems like a critical point for understanding the mechanisms at work. The authors should at least provide model-based predictions regarding the selectivity of the receptor.

Line 216: Increased should be increase

Lies 220-226: According to their location in the neurohypophysis, the AMCA-labeled/TSH-R positive cells seem to be pituicytes rather than FS cells. In any case, the role of glial cells (such as tanycytes) in TSH signaling is well known in mammals and should be discussed (see PMID: 25437536 and PMID: 35361916).

Line 254: mNRA should be mRNA.

Line 310: Change "opposite to in"

Lines 320-329: Reads too much like a literature review rather than a discussion.

Line 356: consider changing "powerful" to potent.

Line 328: Change "to be" to "to being"

Reference 60 and 88 are the same reference.

The difference between the dissociated cells and intact gland is interesting since paracrine factors should be present and functional in both experimental systems. Perhaps it's the structural network that is lacking in the dissociated cells system that is responsible for propagating the signals? Or

maybe it is due to changes that dissociated cells go through during culture as the authors describe in previous publications (PMID: 31977313)? Or the presence of 5% bovine serum in the organ culture?

How do the authors explain the fact that melatonin does not affect dissociated tsh cells in culture? Is the effect also indirect? If so, it should be integrated into the model (figure 7).

In general, the role of melatonin seems unconvincing since it does not affect tshba in females, the sex most affected by photoperiod. Perhaps other factors, such as deep brain opsins, affect the gland?

Did the authors consider a possible effect of TSH through the increase of thyroid hormone production, a known stimulator of gonadotrophs? Even in the cultured pituitary glands, TSH may increase DIO2 expression (see PMID: 25437536 and PMID: 35361916) which further increases T3 levels by converting the T4 available in the 5% fetal bovine serum that was added to the organ culture.

Reviewer #2 (Remarks to the Author):

Authors investigated how photoperiod regulates gonadotropin mRNA production and gonadotrope cell number, as well as the regulatory pathways involved using as model a photoperiodic species, the Japanese medaka. They found that gonadal development is stimulated via increased gonadotropin mRNA production that is partly due to an increase of the gonadotrope cell number. Interestingly, they showed a role of melatonin and tsh in an intra-pituitary pathway that regulates gonadotrope cell activity.

Overall, I think this is an interesting study and I believe that the suggestions below, including additional analyses, would significantly strengthen the manuscript.

My comments that the authors should address are:

- Lines 131-133: Authors suggested that males are less affected by photoperiod for the control of their reproductive activity respect to the female. Indeed, males treated with short photoperiod were still able to reproduce. In natural condition medaka is exposed to short photoperiod and cool-temperature conditions. Shimmura et al. (Nat Commun (2017) 8: 412) showed that winter conditions (10 h light: 14 h dark; 8 °C) altered medaka behaviours including mate preference. They proposed that the downregulation of LWS opsin expression in winter is crucial in the inhibition of this behaviour. Authors did not discuss this paper but the temperature could play a role in the complete inhibition of reproductive activity.

Although authors did not use the male in the main experiments of the paper, they should taking into account also this environmental parameter at least in the discussion.

- Authors confirmed pituitary melatonin receptors are differentially regulated by photoperiod through the regulation of Tsh synthesis and on gonadotrope cell proliferation. They showed a photoperiod-dependent expression levels of mel receptors with higher levels in short photoperiod when compared to long photoperiod fish.

Authors should discuss the reason of the lower levels of mel receptors during the reproductive season (long photoperiod). Is it expected? Is it related to the plasma melatonin levels? How is related to the reproductive activity?

- Authors used two-sample independent t-test or Mann-Whitney U test, but in some tests (e.g., Fig. 1 B-G, Fig. 2C, Fig. S1B) the use of parametric or non-parametric ANOVA (one or two way) and post-hoc analysis are more appropriate. Authors should revise all statistical analyses.

Minor comments:

- Line 122 and 254 correct mNRA with mRNA

- Fig. S1A is not cited in the text

- Line 516 Number of the authorization for use of animal from the Ethical Board is necessary

- Line 384 Information of light sources are necessary (intensity, spectrum, etc)

- Line 403 Is the euthanized methods authorized? EU law did not permit cryoanesthesia for the euthanasia.

Reviewers' comments:

Reviewer #1 (Remarks to the Author):

In this manuscript, Royan and co-authors describe an intra-pituitary paracrine circuit by which thyroid-stimulating hormone (TSH) stimulates gonadotroph cells indirectly through folliculostellate/pituicyte cells. By showing that TSH expression is affected by photoperiod (perhaps by melatonin) they claim that the aforementioned circuit is responsible for the photoperiod-dependent control of reproduction in this species.

The experimental data is very well presented and the TSH-FS-gonadotroph circuit is an interesting and novel finding in fish. However, the link between photoperiod and the circuit is less convincing and based too strongly on correlative observations.

We thank the reviewer for the compliment on our manuscript, especially concerning the novelty of our study in fish. We agree with the reviewer's comments on the fact that some of our claims might be too strong. We have therefore revised the manuscript to tone down several of our conclusions.

We see that melatonin receptors in the pituitary react to photoperiod and show that that melatonin downregulates *tshba* expression. While our study supports an indirect effect of melatonin on TSH cells, we still do not know what cell plays the intermediate role and we do not know how important the melatonin signal for seasonal reproduction in fish is. Both will be particularly interesting topics to investigate in future, and to compare to what have been found in mammals. For this reason, all results regarding melatonin are in the supplemental data and we now revised the result section and carefully discuss the potential role of melatonin by adding more literature on its role. We have also removed it from our model in the last figure and conclusion as it is somewhat speculative.

Specific comments:

Line 42-44: The suggestion that the same mechanism applies to other vertebrates cannot be based just on the presence of the components in other species. Moreover, the role of seasonal TSH signaling through glial cells was described in mammals (PMID: 35361916) so the authors should perhaps rephrase their claim to state that the mechanism also exists in fish.

The paper to which the reviewer refers is presenting what is today very well known in mammals: the effect of TSH produced in the pars tuberalis on photoperiod signal integration via the brain (DIO2 tanycytes). A similar pathway (pituitary Tsh- Brain DIO cells) has now been described in fish as well, as in Atlantic salmon (Irachi et al, 2021). In the present study, we show a pathway which has never been described in any vertebrate: a communication path between TSH cells and the gonadotropes which remains within the pituitary (via pituitary FS cells). As pituitary FS cells expressing Tsh receptors also exist in mammals this is why we believe that the intra-pituitary pathway we describe in fish may also exist in mammals. We anyway removed this statement from the abstract to be able to explain it more clearly in the discussion part (Line 450-453).

Irachi S, Hall DJ, Fleming MS, Maugars G, Björnsson BT, Dufour S, Uchida K, McCormick SD. Photoperiodic regulation of pituitary thyroid-stimulating hormone and brain deiodinase in Atlantic salmon. Mol Cell Endocrinol. 2021 Jan 1;519:111056. doi: 10.1016/j.mce.2020.111056.

Throughout the study, the authors should not use the term cell "activity" as it may be misleading. Activity usually refers to electrical activity which they do not measure. Gene expression is a more appropriate term in this case since this is the parameter they use as a readout.

We thank the reviewer for the comment have revised the term accordingly throughout the manuscript.

Authors should consider including PMID: 21093265 in their intro/discussion as it describes photoperiodic sensitivity of tsh genes in fish.

We adopted reviewer's suggestion in the manuscript by citing the suggested article in the discussion.

Line 125: Consider revising "the double" into 2-fold higher.

We have revised accordingly.

Line 125: Is support for the importance of FSH and LH in reproduction required? Perhaps it provides further evidence for the importance of hyperplasia in the gland's plasticity?

We thank the reviewer for spotting this ambiguity. We have rephrased the sentence.

Lines 151-156: The paragraph makes a statement (*gnrh1* and kiss signaling are elevated in SP toLP transition) and then counters it in the second part (the genes are also elevated in the opposite shift). It is unclear what the readers should make of this.

We apologize for the confusion. We meant that since *gnrh1* and *kiss1* gene expressions respond similarly to both photoperiod transitions while reproductive activity responds differently to different photoperiod transitions. We have modified this statement after discussing it in detail with another neuroendocrinologist. (Line 171-183)

The authors should provide proof or at least cite papers that prove that bovine TSH activates the fish receptors. Considering the low homology of the bovine and medaka proteins (Supp. Figure 4) it is not a trivial question.

We agree with the reviewer that the mammalian TSH β and fish Tsh β are relatively different. In fact, we performed an alignment of the mammalian TSH-receptor and the pituitary medaka Tsh-receptor and found that except for small very well conserved parts, the protein sequence also differs for the receptor (43% identity). Nevertheless, even if both ligand and receptors are quite different between fish and mammals there are several studies showing that mammalian TSH can bind fish Tsh-receptors (e.g. zebrafish (Opitz et al., 2011), channel catfish (Goto-Kazeto et al., 2009), striped bass (Kumar et al., 2000)). We now include and discuss these references in the result (Line 229) section of the manuscript.

Robert Opitz, Emilie Maquet, Maxime Zoenen, Rajesh Dadhich, Sabine Costagliola, TSH Receptor Function Is Required for Normal Thyroid Differentiation in Zebrafish, Molecular Endocrinology, Volume 25, Issue 9, 1 September 2011, Pages 1579–1599, <https://doi.org/10.1210/me.2011-0046>

Goto-Kazeto R, Kazeto Y, Trant JM. Molecular cloning, characterization and expression of thyroid-stimulating hormone receptor in channel catfish. *Gen Comp Endocrinol*. 2009 May;161(3):313-9. doi: 10.1016/j.ygcen.2009.01.009. Epub 2009 Jan 23. PMID: 19523396.

Kumar RS, Ijiri S, Kight K, Swanson P, Dittman A, Alok D, Zohar Y, Trant JM. Cloning and functional expression of a thyrotropin receptor from the gonads of a vertebrate (bony fish): potential thyroid-independent role for thyrotropin in reproduction. *Mol Cell Endocrinol*. 2000 Sep 25;167(1-2):1-9. doi: 10.1016/s0303-7207(00)00304-x. PMID: 11000515.

Line 189: Why is this evidence for direct activation of aromatase-expressing cells? Can they not be activated in a paracrine manner by other cells? Gonadotrophs express aromatase and therefore they can be the source of its rise. Is there an increase of E2 in the medium? That could be a way to induce at least LH expression.

We agree with the reviewer, aromatase is not only expressed in FS cells. Gonadotropes also express aromatase but according to both our present single cell data and in situ hybridization work (Fontaine et al 2019), FS cells express very high levels of aromatase compared to gonadotropes. Nevertheless, at this step of the manuscript we do not speculate on the cell type stimulated. While we did not evaluate the E2 level in the medium, the *ex vivo* culture medium does not contain testosterone thus limiting the production of E2. It is therefore highly unlikely that we have stimulation of spread cells via E2 production. Even if we agree with the reviewer that this is not a direct proof that aromatase expressing cells are directly stimulated by TSH, as we wrote in the manuscript the use of in vitro dissociated cell culture can indicate a direct stimulation.

Although it is not a direct proof of their activation, FS cells are the only one having Tsh receptors. As these cells are also the one expressing the highest level of aromatase, our results support that FS cells are the one stimulated by TSH and that increase aromatase mRNA level in the culture. Because of the manuscript's narrative structure, this association will only be discussed later in the manuscript.

Is the medaka TSHR2 activated by TSHba or TSHbb? Or both? This seems like a critical point for understanding the mechanisms at work. The authors should at least provide model-based predictions regarding the selectivity of the receptor.

We agree with the reviewer that this is very interesting information. Unfortunately, the Tsh β a and Tsh β b have not yet been produced in any fish so far. This should definitely be the next step to be able to investigate their binding affinity to the different receptors. However, as it can be observed in figure S4B, Tsh β a differ quite substantially between fish species, and the difference is even bigger between Tsh β b. This will likely make difficult the comparative study of these two proteins between species and thus certainly require individual protein production and study for each species.

As mammalian TSH is found to activate fish Tsh-receptors and that Tshba is more similar to mammalian than Tshbb, it is likely that Tshba can activate the fish sTsh receptor. As already mentioned in the discussion, nothing is known for Tshbb, unfortunately. Together with other groups, we currently consider the possibility that Tshbb could bind other receptors than Tsh-receptors. But that is only speculation for the moment and further studies are needed to characterize Tshbb.

Line 216: Increased should be increase

Revised.

Lies 220-226: According to their location in the neurohypophysis, the AMCA-labeled/TSH-R positive cells seem to be pituicytes rather than FS cells. In any case, the role of glial cells (such as tanycytes) in TSH signaling is well known in mammals and should be discussed (see PMID: 25437536 and PMID: 35361916).

Whether these FS cells are pituicytes, is unclear for now. Both FS cells and pituicytes have previously been defined based on morphological and histological characteristics. Our single-cell studies on the medaka pituitary provide a complementary characterization. In all these studies, FS cells are a heterogeneous complex; our single-cell studies suggest that transcriptomically, pituicytes are members of this complex. Disentangling this complex at the transcriptional and histological levels is complicated by the fact that both cell types are labeled by the AMCA dipeptide.

Anyway, these cells are clearly pituitary cells and therefore different from the tanycytes which have been described in the mammalian brain.

Line 254: mNRA should be mRNA.

Revised.

Line 310: Change "opposite to in"

Revised.

Lines 320-329: Reads too much like a literature review rather than a discussion.

We thank the reviewer for the comment and have revised the paragraph.

Line 356: consider changing "powerful" to potent.

Revised.

Line 328: Change "to be" to "to being"

Revised.

Reference 60 and 88 are the same reference.

Revised.

The difference between the dissociated cells and intact gland is interesting since paracrine factors should be present and functional in both experimental systems. Perhaps it's the structural network that is lacking in the dissociated cells system that is responsible for propagating the signals? Or maybe it is due

to changes that dissociated cells go through during culture as the authors describe in previous publications (PMID: 31977313)? Or the presence of 5% bovine serum in the organ culture?

The reviewer makes a good point that although the cells are dispersed, the paracrine factors could still be present, even if they are diluted. This is indeed supported by several previous publications mentioned in Francis et al, 1997. We therefore reformulated our statement with some citations to highlight that the main difference between the ex vivo and in vitro is the lack of structural network and the reduction of paracrine signals.

It is also true that some changes in structural network and even cell phenotype (as shown after 24h in culture in our previous article) may affect our results. However, to limit the effect of cell phenotypic change, all in vitro and ex vivo experiments were performed as quickly as possible. The tissues/cells were incubated in the media right after sampling or dissociation, and the cell were collected in trizol just after the 24h treatment. This means that the experiment was done in about 24hours minimizing at its maximum the risk of phenotypic change. We thus revised the material and method parts to provide more info regarding the timing of the experiment and we now mention a sentence about this risk in the discussion.

Finally, we cannot completely rule out that the bovine serum has an effect and therefore added a sentence in the discussion (Line 431-439)

Francis K, Palsson BO. Effective intercellular communication distances are determined by the relative time constants for cyto/chemokine secretion and diffusion. Proc Natl Acad Sci U S A. 1997 Nov 11;94(23):12258-62. doi: 10.1073/pnas.94.23.12258.

How do the authors explain the fact that melatonin does not affect dissociated tsh cells in culture? Is the effect also indirect? If so, it should be integrated into the model (figure 7).

Indeed, we believe that the effect of melatonin on TSH cells is indirect in this case. That is why we made the model with dashed line from melatonin to TSH cells, to indicate that there are some other intermediate factors that mediate the signaling. We have now tried clarifying this in the discussion. (Line 555-560)

In general, the role of melatonin seems unconvincing since it does not affect tshba in females, the sex most affected by photoperiod. Perhaps other factors, such as deep brain opsins, affect the gland?

Although the suppression of *tshba* by melatonin does not reach statistical significance in females, it clearly shows a tendency of suppression. In fact, a previous study in medaka, using the same protocol (PMID: 32517612) showed that melatonin significantly decreased *tshba* levels in female medaka pituitary. We believe that the non-significance in our results is associated with the high individual variation and the lower number of samples (n). We now try to make this clearer in the manuscript.

We agree with the reviewer regarding the role of deep photoreceptors. In fact, the role of circulating melatonin from the pineal is still unclear in fish. Studies of pinealectomy found no effects on reproduction. On the other hand, deep photoreceptors have been described in all tissues in fish, including in the medaka pituitary (Davies et al, 2015). In fact, a new study still in preprint shows that deep photoreceptors such as opsin can play a role in pituitary hormone release

(<https://doi.org/10.1101/2023.08.02.551597>). Opsin should thus receive more attention in the future. We now include this in the discussion. (Line 561-640)

Davies, W.I., Tamai, T.K., Zheng, L., Fu, J.K., Rihel, J., Foster, R.G., Whitmore, D., Hankins, M.W., 2015. An extended family of novel vertebrate photopigments is widely expressed and displays a diversity of function. Genome Res 25(11), 1666-1679

Did the authors consider a possible effect of TSH through the increase of thyroid hormone production, a known stimulator of gonadotrophs? Even in the cultured pituitary glands, TSH may increase DIO2 expression (see PMID: 25437536 and PMID: 35361916) which further increases T3 levels by converting the T4 available in the 5% fetal bovine serum that was added to the organ culture.

The reviewer raises a good point. We checked in our medaka pituitary single cell data and we found almost no cells expressing DIO2. However, DIO1 is expressed in a few pituitary cells. The higher expression levels of DIO1 are found in the gonadotrope cell culture but some expression is found in the FS cells. Tsh-receptors are only found in FS cells which does not seem to be cell group expressing the highest levels of Dio1. In addition, whether the few FS cells expressing DIO1 can convert T4 from fetal bovine serum is unknown. Therefore, while it is possible that FS cells communicate with gonadotropes via T3, we believe that it is too speculative with the current available data to mention it in the manuscript.

Reviewer #2 (Remarks to the Author):

Authors investigated how photoperiod regulates gonadotropin mRNA production and gonadotrope cell number, as well as the regulatory pathways involved using as model a photoperiodic species, the Japanese medaka. They found that gonadal development is stimulated via increased gonadotropin mRNA production that is partly due to an increase of the gonadotrope cell number. Interestingly, they showed a role of melatonin and tsh in an intra-pituitary pathway that regulates gonadotrope cell activity.

Overall, I think this is an interesting study and I believe that the suggestions below, including additional analyses, would significantly strengthen the manuscript.

We thank the reviewer for the positive interest to our manuscript. We believe we addressed all the comments.

My comments that the authors should address are:

- Lines 131-133: Authors suggested that males are less affected by photoperiod for the control of their reproductive activity respect to the female. Indeed, males treated with short photoperiod were still able to reproduce. In natural condition medaka is exposed to short photoperiod and cool-temperature conditions. Shimmura et al. (Nat Commun (2017) 8: 412) showed that winter conditions (10 h light: 14 h dark; 8 °C) altered medaka behaviours including mate preference. They proposed that the downregulation of LWS opsin expression in winter is crucial in the inhibition of this behaviour. Authors did not discuss this paper but the temperature could play a role in the complete inhibition of reproductive activity.

Although authors did not use the male in the main experiments of the paper, they should taking into account also this environmental parameter at least in the discussion.

We would like to highlight to the reviewer that most of the experiments were conducted on both females and males, but the data on males are mostly found in the supplemental data.

We agree with the reviewer that Winter is not only a decrease in photoperiod but also a decrease in temperature. It is possible, that temperature plays an important role in *lhb* and *fshb* synthesis or gonadotrope cell proliferation, in males and in females. We now mention this possibility in the discussion.(Line 363-366)

- Authors confirmed pituitary melatonin receptors are differentially regulated by photoperiod through the regulation of Tsh synthesis and on gonadotrope cell proliferation. They showed a photoperiod-dependent expression levels of mel receptors with higher levels in short photoperiod when compared to long photoperiod fish.

Authors should discuss the reason of the lower levels of mel receptors during the reproductive season (long photoperiod). Is it expected? Is it related to the plasma melatonin levels? How is related to the reproductive activity?

We indeed found the melatonin receptors were down regulated in LP in agreement with previous medaka studies. We believe that because of the lower circulating levels during night in LP conditions, the need of receptors is also lower. However, as discussed with the first reviewer, the role of melatonin in the regulation of reproduction is still unclear in fish. Indeed, several fish species were not affected at the reproductive axis by melatonin changes as for instance in salmon (Mayer, 2000)(for review: Acharyya et al, 2021). We have developed the discussion regarding melatonin role and melatonin receptors levels following both reviewer comments.(Line 555-563)

Ian Mayer. Effect of long-term pinealectomy on growth and precocious maturation in Atlantic salmon, Salmo salar parr. Aquatic Living Resources, Volume 13, Issue 3, 2000

Acharyya, A., Das, J. & Hasan, K. N. Melatonin as a Multipotent Component of Fish Feed: Basic Information for Its Potential Application in Aquaculture. Frontiers in Marine Science 8 (2021). <https://doi.org/10.3389/fmars.2021.734066>

- Authors used two-sample independent t-test or Mann-Whitney U test, but in some tests (e.g., Fig. 1 B-G, Fig. 2C, Fig. S1B) the use of parametric or non-parametric ANOVA (one or two way) and post-hoc analysis are more appropriate. Authors should revise all statistical analyses.

We agree with the reviewer and have now revised the statistics for the data in the above-mentioned figures, except for fig.S1B which is only illustrative (No statistic test has been run on this figure). Fig 1 and Fig 2 have therefore been revised accordingly to include the new statistics.

Minor comments:

- Line 122 and 254 correct mNRA with mRNA

Revised.

- Fig. S1A is not cited in the text

We adopted reviewer suggestion and we now cite Fig. S1A in the text.

- Line 516 Number of the authorization for use of animal from the Ethical Board is necessary

This is now provided in the manuscript and it covers the use of ice water to euthanize the fish and brood incubation.

- Line 384 Information of light sources are necessary (intensity, spectrum, etc)

We now provide this information in the supplemental data.

- Line 403 Is the euthanized methods authorized? EU law did not permit cryoanesthesia for the euthanasia.

The reviewer is right, this technique is not yet allowed by EU. In fact the EU is currently changing this rule as ice water has been shown to be the most humane way to euthanize zebrafish and medaka which are tropical species. Ice water for euthanasia was approved in our special permits from the Norwegian authorities FOTS id 24305. The number of the permit now appears in the manuscript.

REVIEWERS' COMMENTS:

Reviewer #1 (Remarks to the Author):

I find that the authors sufficiently addressed most of my comments. Where the comments have not been fully addressed, I find that the replies provided detailed answers and added the required explanations and clarifications in the revised manuscript.

I therefore recommend to accept the manuscript.

Reviewer #2 (Remarks to the Author):

Authors replied to all my comments and, in my opinion, the paper has been significantly improved.